# Imaging of Canine Neoplastic Reproductive Disorders

**DOI:** 10.3390/ani11051213

**Published:** 2021-04-22

**Authors:** Marco Russo, Gary C.W. England, Giuseppe Catone, Gabriele Marino

**Affiliations:** 1Department of Veterinary Medicine and Animal Production, University of Naples, Federico II, 80137 Naples, Italy; 2Sutton Bonington Campus, School of Veterinary Medicine and Science, University of Nottingham, Loughborough LE12 5RD, UK; gary.england@nottingham.ac.uk; 3Department of Veterinary Sciences, University of Messina, 98168 Messina, Italy; giuseppe.catone@unime.it

**Keywords:** dog, ovaries, uterus, vagina, testes, prostate, penis, radiography, ultrasound, computed tomography, magnetic resonance

## Abstract

**Simple Summary:**

The diagnosis of canine reproductive neoplasia remains challenging as none of the routinely performed diagnostic methods appear to have sufficient sensitivity or specificity. In recent years, advanced imaging techniques have been successfully performed in small animals; however, even though the incidence of reproductive neoplasia is high, no data are available on the performance of these techniques. This review evaluates the applicability of various diagnostic imaging modalities in dogs and describes the findings and specific patterns that may characterise different tumour types. Lamentably, some of the advanced imaging techniques have not yet been adopted as first-line diagnostic tools, although it is clear that in the future they will become important methods for the detection of male and female reproductive neoplasia.

**Abstract:**

Diagnostic imaging plays an essential role in the diagnosis and management of reproductive neoplasia in dogs and cats. The initial diagnosis, staging, and planning of surgical and radiation treatment and the response to therapy all involve imaging to varying degrees. Routine radiographs, ultrasound, nuclear medicine, and cross-sectional imaging in the form of computed tomography (CT) and magnetic resonance imaging (MRI) are routinely used in canine reproductive disorders. The choice of imaging modality depends on many factors, including the level of referral and the pathological information required. The biological behaviour of the tumour also guides the choice of imaging in cancer staging, and imaging may play an important role in guiding serial tumour biopsy during the course of therapy. The sophistication of imaging modalities is increasing exponentially. Each modality has advantages and disadvantages in terms of cost, availability, sensitivity, specificity, and qualities of anatomic versus functional imaging.

## 1. Introduction

Several imaging modalities are now available in small animal oncology and in the study of neoplasia involving the female and male reproductive tracts. Each modality has advantages and disadvantages that may relate to cost, availability, sensitivity, and specificity; some provide ‘anatomic imaging’ whilst others provide ‘functional imaging’. Each method may have different roles in diagnosis, staging, treatment selection and follow-up. The use of diagnostic imaging as an aid in canine and feline reproduction has evolved over the past twenty years from its initial role in early pregnancy diagnosis to its current use as an integral component in the management of reproductive cases. Ultrasonography is sensitive for the detection of lesions, but it is not specific for the aetiology of a disease. Therefore, biopsy or fine-needle aspiration of the lesion may be necessary. Ultrasound-guided sampling of tissue can be performed quickly, accurately, and safely. Advances in ultrasound equipment and the development of ultrasonographic contrast agents are increasing the diagnostic specificity of ultrasound. For example, Doppler techniques are used to detect tumour vessels, which are usually seen to be tortuous and at high velocity compared with vessels present within normal tissue.

This review describes both standard and innovative approaches to imaging in dogs with suspected reproductive malignancy and highlights the important contribution of imaging to the management of these patients.

## 2. Female

### 2.1. Reproductive Anatomy

The ovaries are located immediately caudal to the kidneys and are positioned close to the abdominal wall. The uterus of the bitch is roughly ‘Y’-shaped and has relatively long uterine horns which are positioned proximally, close to the abdominal wall, converging distally in the midline at the uterine body.

### 2.2. Imaging of the Ovaries

The ovaries are located in the dorsal abdomen slightly lateral to the caudal poles of the kidneys. Normal-sized or mildly enlarged ovaries are not visible radiographically, limiting the usefulness of this technique and making ultrasound the first-choice in the evaluation of ovarian disorders. Multifrequency curvilinear or linear 5–11 MHz transducers are the standard equipment for ovarian examination. High-resolution probes (i.e., 18 MHz or more) are currently available and may maximise the ability to visualise subtle changes in the ovary. The caudal pole of the kidneys and the adjacent area are examined in sagittal and transverse planes to localise the ovaries. Their identification may be facilitated by the appearance of marginal artefacts dorsal to each ovary. The changes seen in the bitch ovary during the reproductive cycle have been described [1]. The appearance varies according to the stage of the oestrous cycle when follicular growth can be readily detected. Multiple anechoic structures can be observed during proestrus and oestrus; after ovulation, thicker-walled corpora lutea are present during late oestrus and the first phase of dioestrus. Round and hypoechoic corpora lutea are easily detectable in mid-dioestrus when they deform the otherwise oval profile of the ovary.

#### Imaging of Ovarian Neoplasia

Survey radiography was the first technique described for imaging canine ovarian tumours; however, radiography provides little information regarding ovarian mass architecture. When the ovary is significantly enlarged, it results in a ventrally located mass that causes medial, but not ventral displacement of adjacent organs on radiographs. Right ovarian masses displace the descending duodenum and ascending colon medially, whereas left ovarian masses displace the descending colon medially [2,3,4,5,6,7,8,9,10,11,12,13]. Peritoneal or pleural effusion may be present when there are either benign or malignant masses [2,13]. Distant metastases may be detected in thoracic projections [6]. Radiographic examination is not sensitive and other tumours, atypical pyometra, and retroperitoneal abscesses cannot be excluded. Occasionally, ovarian teratomas and teratocarcinomas contain mineralised areas, which are visible radiographically [4]. The easy accessibility and relative low cost of ultrasound have made it the study of choice in the initial evaluation of a patient with a suspected ovarian neoplasia.

Confirmation that a mass is ovarian in origin is based on its appropriate location, being caudal to the kidneys, and ideally having an association with an adjacent uterine horn. The adjacent uterus can be confirmed by imaging the ovarian veins with Colour Doppler. Ultrasonographically, ovaries in bitches with ovarian tumours may appear unilaterally enlarged, with regional or focal lesions that may be solid or cystic. Frequently there is a significant disruption of the normal appearance with an inhomogeneous echotexture. Tumours may be small or large and may be solid, contain small cysts, or be primarily cystic in appearance. A literature search of ultrasound imaging of ovarian masses identified only two case series [2,13] and numerous case reports. Ultrasonography appears to be sensitive for the detection of ovarian masses, but there are no exclusive patterns that confirm the diagnosis of tumour type. Unfortunately, no morphological scoring systems is available in veterinary medicine to standardise the interpretation of ultrasound images.

An attempt to classify ovarian masses based on ultrasonographic appearance has been proposed [2] and includes three groups: solid masses (less than 10% anechoic cavities), solid masses with a cystic component (from 10% to 50%), or cystic masses (greater than 50%) [2]. Ultrasound images of the ovarian neoplasia were described on their location, size, outer margins, and echogenicity presence of free abdominal fluid, evidence of uterine abnormalities, and signs of metastatic disease. The tumours were ultrasonographically classified as solid (adenocarcinoma, thecoma) (solid with cystic component (adenocarcinomas, granulosa cell tumour, dysgerminoma)), and cystic (adenoma, teratoma). The size of the anechoic cavities ranged from 0.2 to 3.5 cm in diameter. The solid parenchyma had a fairly uniform appearance in all [5,7,8]. Teratomas are often cystic and may show partial mineralisation with distal shadowing, due to structures such as hair, skin, sweat glands, cartilage, bone, and teeth, which might help to distinguish these tumours from other ovarian masses. Finally, other rare tumours within the ovary have been described and imaged, including leiomyoma, which typically has a solid appearance [12] and hemangiosarcoma, which has been reported as solid with a heterogeneous echotexture. Uterine changes (cystic endometrial hyperplasia or pyometra) and/or ascites are commonly seen in hormonally active (sex-cord stromal) tumours [2] but are not exclusive to such tumours [7]; the presence of free fluid may be associated with peritoneal dissemination, and careful evaluation for metastatic disease should be performed. Colour Doppler ultrasound may help identify solid, vascularised components in an ovarian mass [9]. In humans, spectral Doppler waveform characteristics were shown to correlate well with malignancy [13]. Furthermore, contrast-enhanced ultrasound (CEUS) has been used for the pre-operative evaluation of early benign or malignant masses by imaging tumour microvascularity [14]. Unfortunately, there are no studies describing the use of these techniques in canine ovarian tumours.

When imaged with computed tomography (CT), ovarian tumours in dogs appear as large soft-tissue attenuated masses located in the mid-ventral abdomen, with moderate or severe contrast enhancement [9]. CT may help to assess the extent of disease in patients before and after primary surgery. The separation of abdominal organs from the tumour may be easily visualised (Figure 1A,B).

With CT, the origin of the tumour may be confirmed, and the assessment of surgical options may be significantly improved compared with other imaging modalities. Similar to radiography, CT may identify mineralised areas in some ovarian masses and tooth-like structures in teratomas. The CT appearance of an ovarian papillary adenocarcinoma has been described, showing an obvious internal structure, areas of contrast enhancement. and connection with the ovarian vein [9]. An ovarian dysgerminoma associated with pleural and abdominal effusions was also detected by CT and followed during chemotherapy [5]. Ovarian leiomyoma has been described as a heterogeneous mass with intense contrast enhancement [12]. Magnetic resonance imaging (MRI) imaging is another advanced imaging technique that may be used when ultrasound findings are non-diagnostic or equivocal. It is especially useful to investigate distant metastasis in specific organs.

### 2.3. Imaging of the Uterine Tube and Uterine Tube Neoplasia

The function of the canine uterine tube, also called the oviduct (or the Fallopian tube), is to carry the ova from the ovary to the uterine horn. Each Fallopian tube is a narrow structure that lies near the ovary and passes over it into the uterine horn. They are not visible with modern imaging techniques unless they enlarge due to fluid accumulation or neoplasia. Tumours are extremely rare and are unlikely to become symptomatic until they reach a large size. At that stage, uterine tube masses may be seen on radiographs as a generic soft tissue mass. Ultrasound is a more sensitive technique than radiography, and it is possible to speculate that mass lesions adjacent to the ovary may originate from the uterine tube. Leiomyoma of the mesosalpinx has been described as a solid mass [15]. Large adenomas have been described as unilocular cystic or cavernous masses [16,17]. In one case, an adenocarcinoma was described as an ovoid mass with a moderately vascularised heterogenous parenchyma and hypoechoic cystic areas. Two of the cited case reports were also studied by CT to reveal their internal structure and topography [17].

### 2.4. Imaging of the Uterus

Methods of uterine imaging include survey radiography, ultrasound, CT, and MRI. As described under ovarian mass lesions, the main limitations of survey radiography are subject density and size. The uterus has the same subject density as adjacent soft tissue structures, and, therefore, it cannot be identified radiographically. Ultrasound imaging of the uterus is best performed with the bitch in a standing position after clipping the hair of the ventral abdomen. Partial filling of the bladder helps identify the body of the uterus and the larger diameter of the cervix. The uterine bifurcation can be detected in about 40% of cases. The uterus is surprisingly tortuous and may coil and position itself in unexpected directions, but the proximal portions of the two uterine horns can be imaged and traced laterally to the abdominal wall. They often lie along the length of the body wall. The uterine body and horns are composed of two distinct layers: a central homogeneous, relatively hypoechoic region surrounded by a peripheral hyperechoic layer. The ability to differentiate these layers depends upon the stage of the cycle. During oestrus, the uterus becomes increasingly hypoechoic and is much larger in diameter. The lumen is generally not seen but may be visible as a bright echogenic central line, which represents the mucosal–luminal interfaces. During proestrus and oestrus, there may be minimal anechoic content (fluid) 1 mm wide in the lumen [1].

#### Imaging of Uterine Neoplasia

Uterine neoplasia is rare in the bitch, and since imaging techniques cannot readily distinguish neoplastic from granulomatous diseases, fine-needle aspiration is often required to make a definitive diagnosis. Diagnostic imaging is an important step in identifying uterine abdominal masses and subsequent treatment planning. However, the size and nature of large uterine masses can make interpretation of abdominal radiographs and ultrasound challenging. Two retrospective studies [18,19] and many case reports are available in the veterinary literature. Large uterine masses, despite their nature, have been described radiographically as homogeneous mid-ventral abdominal soft-tissue mass lesions that may displace the small bowel, stomach, and liver [19,20,21,22]. Foci of calcification have been seen in leiomyosarcoma and leiomyoma with dystrophic calcification and osseous metaplasia [19,21]. Soft-tissue opacities greater than 4 cm in diameter were noted between the urinary bladder and colon in six cases with tumours of the uterine body [19]. In half of these dogs, the outer margin of the lesion was well-defined. Cranial displacement of the urinary bladder was observed in four dogs and dorsal displacement of the descending colon in six. The same appearance has also been documented in uterine stump adenocarcinoma [23]. Despite this useful information, radiography has poor sensitivity and segmental pyometra, stump pyometra, early pregnancy, other tumours, cysts, or granuloma may appear similar on survey radiographs and should be included in the differential diagnosis.

Ultrasonography has been described as a useful method to demonstrate the uterine origin of an abdominal mass by tracing the abnormal area to a region of normal appearing uterine horn or body. However, when a large mass is present, identification of the affected organ can be very challenging. Ultrasonographically, uterine neoplasia may appear as endoluminal or intramural, homogeneous or heterogeneous, mass lesions with a possible amount of uterine fluid. Leiomyoma, fibroleiomyoma, and leiomyosarcoma have been described as solid masses, but anechoic ischemic cavities are often observed in the mass, giving a mixed to cystic pattern [19,21] (Figure 2). Occasionally, hyperechoic foci have been reported and are thought to be a sign of calcification, fibrosis, or metaplasia found in this category of tumours. Poorly differentiated sarcomas have been described as solid masses. Adenocarcinomas are rare and occasionally present as masses of mixed echogenicity with solid areas, hyperechoic foci, and cystic components [19,23]. Endometrial polyps have been reported as endoluminal, projecting, well-demarcated masses, solid or with anechoic multiple cystic glands surrounded by a large amount of luminal fluid [18,24,25]. Despite some distinguishable patterns, most masses are described as solid with cystic areas, making diagnosis difficult unless biopsy samples are collected.

Understanding the appearance of the uterus in CT and magnetic resonance (MR) images may be important as the use of advanced imaging becomes more common in the examination of the canine abdomen. Cross-sectional imaging may also help to assess more complex uterine anatomic abnormalities and mass lesions that cannot be clearly confirmed as uterine by other imaging modalities. CT was undertaken to determine the uterine nature of a cystic mass occupying most of the abdomen and was subsequently diagnosed as a lipoleiomyoma and also to reveal another solid uterine leiomyoma [22]. The first was a large, fluid-filled mass with soft tissue septa, whilst the second was a small soft tissue mass that showed contrast enhancement and arose from the cranial uterine horn [22]. A rare case of uterine hemangiosarcoma of the uterine remnant has also been detected by contrast-enhanced CT [26]. The contrast uptake was poor and heterogeneously distributed, as is quite typical for hemangiosarcoma in dogs. Recently, the CT characteristics of uterine/vaginal mesenchymal tumours have been reported in a retrospective case series [27]. Interestingly, malignant tumours measured longer than benign forms and tended to occupy a larger portion of the pelvic canal. Finally, bone involvement was only observed with malignancy.

### 2.5. Imaging of the Vagina and Vaginal Neoplasia

The vagina has mainly a pelvic position. Survey radiographs and ultrasound are useful as screening tools but have limited diagnostic value. Vaginoscopy is very useful for visualisation of deep proliferative luminal lesions, although some large and obstructive masses cannot be easily evaluated. On abdominal radiographs, leiomyomas may be seen as ill-defined soft tissue masses in the ventral pelvic canal, causing dorsal displacement of the rectum [28]. Vaginal masses may extend to the perineum and be pedunculated [29]. Negative or positive vagino-urethrograms may be useful to detect lesions involving the vagina.

Ultrasonographically, it is extremely difficult to image the pelvic canal via a transabdominal approach. Some proximal vaginal masses may be detected by pointing the ultrasound probe caudally to visualise the region immediately caudal to the cervix. It is also feasible to infuse saline into the vagina, thus providing anechoic contrast that can be used to image some proximal vaginal wall lesions. It is also difficult to image the vagina with a perineal transcutaneous approach. Although not usually performed in small animals, transrectal ultrasound has been used to image a vaginal mass [28]. Leiomyomas, which are the most common vaginal neoplasia, may be visible as solid masses, moderately vascularised, with hyperechoic structures suggestive of necrosis or anechoic cavities [28,29].

CT and MR are important imaging modalities for the evaluation of vaginal abnormalities. CT allows localisation of the anatomic site of origin due to its multiplanar imaging capabilities and improved contrast resolution. A CT of the caudal abdomen and perineum allowed assessment of the size and location of vaginal leiomyoma and fibroma [28,29]. Such tumours often appear as soft-tissue masses, moderately and heterogeneously contrast enhanced. Vagino-urethrography is another imaging modality for the evaluation of the vagina and is classically performed by radiography, as mentioned above, or using CT. Air or diluted iodinated contrast agents are administered into the vestibule after Foley catheter placement. Positive contrast may be enhanced by simultaneous intravenous administration of the contrast agent. Vagino-urethrography has been performed in a case of vaginal leiomyoma, readily visualised as a contrast-filled defect [28]. Unfortunately, MRI has rarely been documented in veterinary patients to characterise vaginal masses. Only one MRI description of a large intrapelvic leiomyoma compressing the bladder and rectum has been reported [30].

## 3. Male

### 3.1. Imaging of the Prostate

The prostate is an ovoid-shaped bilobed gland located at the bladder neck, surrounding the proximal urethra. The normal prostate is located within or immediately cranial to the pelvis and is bordered dorsally by the rectum. The gland is smoothly marginated, and ultrasonographically the parenchyma has a moderate homogenous echogenicity in entire dogs, with a fine to medium echotexture.

Prostate size is often easily assessed by measuring the maximum total prostate width from images in the transverse plane, but is better assessed by calculating prostate volume using the formula for the volume of an ellipse:Prostate volume = Length × Width × Height × 0.523.(1)

Prostatic size varies for dogs of different breeds.

#### Imaging of Prostatic Neoplasia

Diagnosis of malignant prostate disease is challenging because benign and malignant lesions may have similar clinical presentations and imaging appearance at the initial stages. The search for improved diagnostic techniques continues, and a variety of other imaging modalities have been reported in human medicine, including CT, MRI, positron emission tomography, and single-photon emission computed tomography [31]. In dogs, prostatic carcinoma is a highly malignant neoplasia with a prevalence of 0.2–0.6% [32]. Prostate neoplasia must be distinguished from non-malignant prostatic diseases in order to begin an appropriate treatment. One study reported that prostatic mineralisation identified with either abdominal radiographs or ultrasound may be considered highly suspicious for prostatic neoplasia, especially in neutered dogs [33,34] (Figure 3).

Ultrasonography is the technique of choice for the evaluation of the canine prostate gland, considering that the plethora of imaging modalities commonly performed in human medicine are not frequently used in dogs. Unfortunately, despite providing excellent images, it may be difficult to differentiate between benign and malignant canine prostatic diseases with ultrasound due to their similar appearance when using this technique [35]. Prostatic neoplasia has a variable B-mode ultrasonographic appearance in both entire and neutered dogs. Early cases of neoplasia are often focal, hypoechoic lesions, which are usually difficult to distinguish from other pathologies, including the patchy appearance seen in cases of benign prostatic hyperplasia. Later in the course of the disease, the parenchymal changes are often diffuse, the gland is not symmetrical, and the margins become irregular. A mixed, hyperechoic, heterogenous parenchyma is often detected, with multiple irregular anechoic regions and zones of calcification generating acoustic shadowing. Medial iliac lymph node enlargement may be marked in these advanced cases.

Recently, CEUS has been performed in dogs, demonstrating this to be an innovative technique capable of quantifying vascular perfusion, essentially by measuring peak flow and transit time within tissues using video-densitometric analysis of real-time images. This technique offers significant advantages over simply measuring flow within organ-supplying vessels and enables targeted specific measurement in suspected lesions. The use of CEUS has been described in both normal dogs and those with a variety of prostatic diseases [36,37]. Prostatic tumours can be detected with CEUS, and there are trends in perfusion parameters between tumour types. Peak perfusion intensity values were found to be higher in prostatic carcinomas than in leiomyosarcomas, and the time to reach peak intensity was faster in the former compared with the latter. In addition, different features can be observed during the wash-in and wash-out phases. For example, in cases of prostatic carcinoma, there is hyper-perfusion of the tumour during the wash-in phase and hypo-perfusion during the wash-out phase when compared to normal dogs. Leiomyosarcomas can be characterised in all phases by a homogenous, anechoic, non-perfused area surrounded by highly vascularised parenchyma.

Elastography is a diagnostic tool that may be used to obtain highly detailed images attributable to changes in tissue stiffness. Elastography has been used to evaluate both the normal prostate gland and testicular disease, but there are no studies about the application of elastography to diagnose prostate tumours in dogs.

In general, CT imaging of abdominal organs has several advantages: organs can be imaged without superimposition; small structures can be identified due to high-resolution; organ size and shape can be assessed in multiple planes with image reconstruction. CT examination is considered a helpful tool for the evaluation of the canine prostate gland [38]. However, few studies have used CT to further investigate its diagnostic benefits for examining the prostate in dogs [39], and none of them describe the features of prostatic neoplasia.

The use of MR to evaluate the prostate is still limited and only one short communication describes the use of MR in canine prostatic tumours [40]. The authors described that the enhancement pattern of prostate lesion, relative contrast enhancement indices (RCEI), and apparent diffusion coefficient (ADC) values of prostate lesions may help to detect prostate adenocarcinoma.

### 3.2. Imaging of the Testes

Radiographs are rarely performed in the evaluation of intra-scrotal testicular diseases. Enlarged testicles or testicular masses may appear as an enlarged scrotal silhouette, but this cannot be distinguished from intra-scrotal fluid or thickening of the scrotal soft tissue contents. Ultrasound examination is a sensitive method for evaluating testicular parenchymal diseases, allowing differentiation of testicular from extra-testicular causes of scrotal enlargement [41]. Improved diagnostic techniques and frequent imaging of the testes in infertility have drawn attention to a significant number of small, solid, and often non-palpable tumours, whose diagnosis and management results significantly problematic for the theriogenologists [42]. Ultrasound images can be obtained in the non-sedated dog in a lateral recumbency or standing position. Clipping of the scrotal hair should be avoided as it often leads to excessive licking and subsequent development of scrotal dermatitis. Instead, copious amounts of ultrasound gel should be used. The testes are imaged in longitudinal, transverse, and dorsal planes. The normal testicular parenchyma appears relatively hypoechoic in echotexture with regular diffuse echogenic stippling scattered evenly throughout the organ [43]. The stippling represents an extension of the fibrous mediastinum, which is responsible for supporting the parenchymal tissue. The mediastinum testis, a fibrous invagination of the tunica albuginea, is located centrally within the testis. In the sagittal plane, this structure appears as an echogenic line approximately 2 mm wide extending from the cranial to the caudal pole, whereas in the transverse plane it appears as a central echogenic circular structure. The epididymis appears hypoechoic compared to the testicular parenchyma. Focal testicular lesions are relatively easy to identify and usually, there is a good relationship with gross pathology findings. Because of the high diagnostic validity of ultrasound, there are no descriptions about the use of CT or MR for the evaluation of dog testes.

#### Imaging of Testicular Tumours

Testicular tumours are the most common tumours of the canine male genitalia and account for approximately 90% of all cancers of the male reproductive tract. Testicular tumours are the second most common tumour affecting male dogs. They may be benign or malignant and may or may not be endocrinologically active. While B-mode ultrasound is extremely useful for detecting testicular tumours, the ultrasonographic appearance of lesions varies and is not specific for the type of tumour [41]. Testicular tumours can range from circumscribed small nodules to large complex masses with heterogeneous echo-pattern and disruption of normal anatomy. At the time of diagnosis, Sertoli cell tumours and seminomas are usually large with mixed echogenicity, resulting sometimes in generalised testicular enlargement. Cryptorchidism predisposes to neoplasia, and the most common tumours are Sertoli cell tumours and seminomas [44,45,46] (Figure 4).

Interstitial cell tumours may appear as well-defined focal hypoechoic lesions. However, areas of haemorrhage and necrosis may occur in all tumour types and may be seen ultrasonographically as disorganised hyperechoic and hypoechoic regions. Other findings that may be associated with testicular neoplasia include areas of calcification within the testicular parenchyma that appear as hyperechoic foci producing acoustic shadowing. There are few descriptions of testicular blood flow in abnormal testes. Colour Doppler ultrasonography shows an increase in blood flow within and around most tumours [47]. While this is useful for tumour detection, the changes noted are not specific for tumour type.

CEUS is a feasible diagnostic tool in the evaluation and detection of testicular lesions, with hyper-enhancement being an important feature in the diagnosis of malignancy. The study published by Volta et al. [48] reported that testes with inhomogeneous parenchyma and a hyper-enhancing pattern were associated with neoplasia (sensitivity: 87.5%, specificity: 100%). Lesions with persistent inner vessels and a hypo-to-isoechoic background were significantly associated with seminomas (sensitivity: 77.8%, specificity: 100%). Testes with non-neoplastic lesions were characterised by a scant/moderate homogeneous enhancement (Figure 5).

### 3.3. Imaging of the Penis

The penis is composed of three principal divisions: the root, the body, and the glans. The root of the penis is composed of the two crura and the bulb of the penis. The body is primarily comprised of the two adjacent corpora cavernosa. The glans is subdivided into a bulbus glandis and a pars longa glandis. The os penis, or baculum, is a feature always present in the male dog. It forms from a paired ossification centre in the corpora cavernosa. On X-ray, the penis is best studied on the lateral view when the os penis is visualised, while the root, body, and glands appear as a uniform soft tissue opacity, ventral to the abdominal wall. The penile anatomy is easily demonstrated by conventional B-mode ultrasonography and can be a useful alternative method for penile assessment whenever its exposition becomes impossible.

#### Imaging of Penile Neoplasia

Penile tumours are relatively rare in domestic animals. The most frequently reported canine penile tumours are transmissible venereal tumour, fibropapilloma, and epithelial tumours, such as squamous papilloma and squamous cell carcinoma [49]. The literature on diagnostic imaging of penile tumours in dogs is sparse. One case report described the radiographic and ultrasonographic appearance of a penile hemangiosarcoma [50]. In particular, lysis of the proximal one-third of the os penis was diagnosed on abdominal radiographs, while a positive contrast urethrography revealed a smoothly marginated filling defect along the dorsal aspect of the urethra at the level of the radiographically observed osteolysis. Ultrasound examination revealed an echogenic mass with a severely irregular and discontinuous periosteal surface of the os penis [51]. Penile ultrasound including colour or pulse-wave Doppler and grey-scale analysis is a valuable diagnostic tool in human medicine. Few studies reported the suitability of B-mode ultrasonography to identify physiological anatomical structures of the canine penis; however, its use in canine andrology has not been established yet.

## Figures and Tables

**Figure 1 animals-11-01213-f001:**
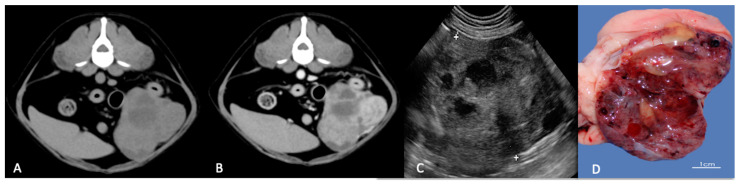
The computed tomography (CT) appearance of a granulosa cell tumour that appears as a large, well-defined low-attenuation ovarian mass. (**A**) Non-enhanced CT scan shows multi cystic soft-tissue mass. (**B**) After contrast administration, CT scan shows mass as mildly and non-homogeneously enhanced. (**C**) Ultrasound longitudinal right ovarian mass with heterogeneous echotexture and multiloculated solid and cystic mass. (**D**) Sagittal cut of the right ovary containing polycystic structures.

**Figure 2 animals-11-01213-f002:**
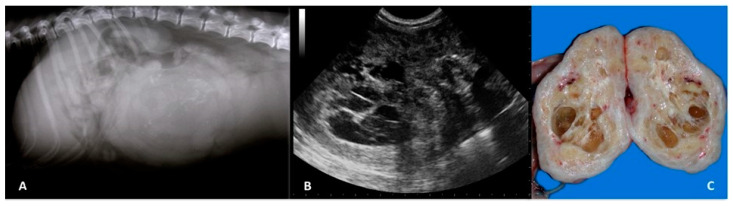
(**A**) Right lateral abdominal radiograph of an 11-year-old pointer female dog with a rounded, ill-defined soft-tissue opacity with mineralised areas at the level of midventral abdomen, displacing the small intestine cranially and the colon dorsally. (**B**) The sagittal ultrasound image shows an inhomogeneous echotexture with multiple, irregular hypoechoic areas with echogenic foci, representing dystrophic mineralisation. (**C**) Gross features of uterine leiomyoma in which multiple large and small cavities are visible representing areas of necrosis and haemorrhage.

**Figure 3 animals-11-01213-f003:**
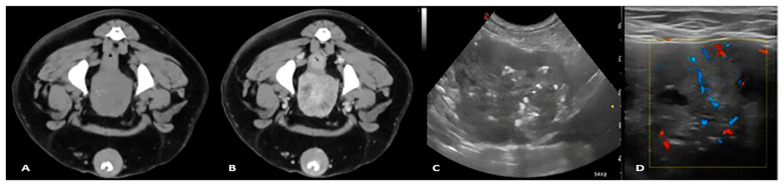
Pre and postcontrast CT study (**A**,**B**), transverse US image, (**C**) and Colour Doppler image (**D**) of a 9-year-old mixed breed dog with prostatic carcinoma. The precontrast study shows fine mineralisation of the prostatic parenchyma; the postcontrast study shows a severely heterogeneous prostate with inhomogeneous contrast enhancement. The B-mode image shows that the prostatic gland has an irregular margin and heterogeneous echotexture with scattered echogenic foci and irregular hypoechoic cyst-like lesions. The Colour Doppler shows irregular branching of arterial vessels throughout the prostatic parenchyma.

**Figure 4 animals-11-01213-f004:**
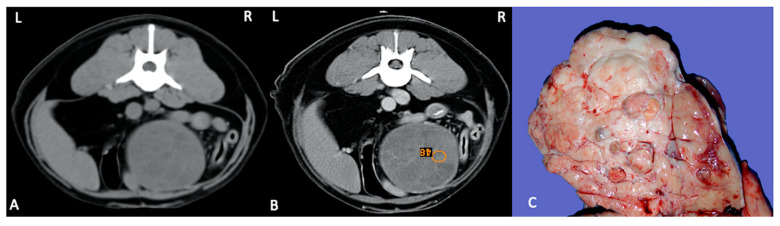
Pre (**A**) and postcontrast (**B**) CT study in dorsal recumbency of a 9-year-old, male entire English Setter with a large well capsulated mass in the mid abdomen with peripheral enhancement and several septae. (**B**) ROI is placed in the mass, and a mild contrast enhancement is visible. Final diagnosis: right testicular cryptorchid Sertoli cell tumour. (**C**) Gross appearance of the cryptorchid Sertoli cell tumour.

**Figure 5 animals-11-01213-f005:**
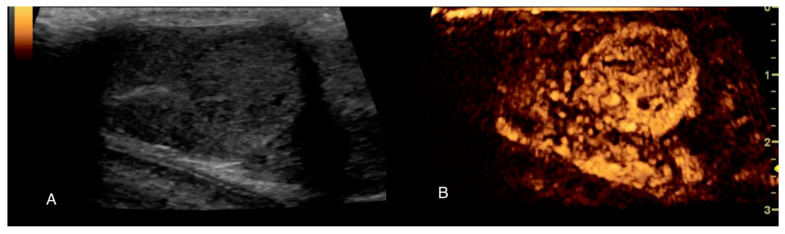
B-mode and Contrast ultrasound examination of the left testis of a 5-year-old mixed breed dog with seminoma. (**A**) B-mode image of the lesion; (**B**) early wash-in with tortuous feeding vessels, 25 s after contrast agent injection. CEUS enabled better and more detailed observation of the microcirculation.

## Data Availability

The data presented in this study are available on request from the corresponding author.

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
