# Peer review of "Imaging of Canine Neoplastic Reproductive Disorders"

_animals, 2021, doi:10.3390/ani11051213_

Round 1

Reviewer 1 Report

The review paper “Imaging of Canine Neoplastic Reproductive Disorders” shows the use of various imaging modalities to diagnose tumors of the female and male reproductive tract. Various newer techniques such as CEUS and Elastography are included. It is shown, that CT and MRI are only used or useful in few cases. There are comprehensive references to each imaging modality and tumor type. The section on ovarian tumors should be re-organized, as it is difficult to follow the various imaging and tumor types. To better illustrate the various techniques and imaging characteristics of reproductive tumors, more figures would be beneficiary to the paper.
General comments:
• The English text is good; there are some suggestions in Specific comments, which may give a better understanding of some of the points.
• The placement of commas should be reviewed, as this also makes reading easier; note the Oxford (serial) comma. I inserted and deleted some commas for better understanding, but not all.
• The headings in the sections are inconsistent. 2. Female continues with 2.1. Reproductive anatomy. In 3. Male, the review starts with 3.1. Imaging of the prostate. For more consistency, there should be a section on reproductive anatomy.
• Also in the headings some consecutive headings begin with imaging of the normal uterus or prostate, with a subheading of imaging of neoplasia, others with imaging of vaginal tumors. Imaging of the penis is actually only imaging of penile neoplasia. Again this should be consistent with a normal imaging section and what is possible, followed by neoplasia imaging.
• In the female section, the headings should be imaging of uterine or ovarian neoplasia, not mass lesions as this would include cysts. In the male section neoplasia was used.
• Figure 2 A and B CT images should be flipped with the spinal cord dorsally.
Specific comments:
Some suggestions to change some of the text, so it is easier to understand.
Line 17 that in the future they will become (no comma)
Line 37 whilst others provide…
Line 56 are positioned close to the abdominal wall proximally
Line 67 High-resolution probes
Line 78 2.2.1. Imaging of ovarian neoplasia
Line 85 …there are either benign or malignant masses
Line 126/127 by imaging tumour microvascularity
Line 138 Figure 1 Sagittal cut
Line 147 mass with intense contrast enhancement
Line 148 MR; see line 23: MRI – please use consistent abbreviations
Line 151 Imaging of uterine tube neoplasia
Line 152 fallopian tube is not capitalized (see line 153)
Line 162 was described as an ovoid mass – no comma
Line 165 uterus not capitalized
Line 166 MR or MRI?
Line 169 structures, and, therefore, it cannot be identified…(insert commas)
Line 170 the bitch in a standing position
Line 170/171 after clipping the hair on the ventral abdomen
Line 178 to differentiate these layers depends…
Line 178 During oestrus, (insert comma)
Line 180/181 which represents the mucosal – luminal interface.
Line 181/182 there may be minimal anechoic content (fluid) 1mm wide in the lumen
Line 183 2.4.1. Imaging of uterine neoplasia
Line 190 in veterinary literature
Line 195 tumours of the uterine body
Line 195/196 In half of these dogs, the outer margin of the lesion was well-defined.
Line 197 four dogs and dorsal displacement (no comma)
Line 206 in the mass, giving a mixed (insert comma)
Line 207 reported and thought (no comma)
Line 210 of mixed echogenicity with solid areas (no comma)
Line 211/212 what is meant by “projecting”? and projecting where?
Line 225 as a lipoleiomyoma and also to reveal another (no comma)
Line 219-230 In this section on CT of uterine tumors, it should be noted that there are only 2 case reports. Are there any more?
Line 231 Imaging of vaginal neoplasia
Line 234 for visualization of deep proliferative
Line 240 Ultrasonographically, it is extremely difficult
Line 259 vaginal leiomyoma, readily visualized as a contrast-filled defect
Line 280 MRI/MR – inconsistent use of abbreviation
Line 284 One study reported that prostatic - delete reference, already at the end of citation
Line 297 with multiple
Line 299 in these advanced cases.
Line 312 hypo-echogenicity – I believe you meant to say ”hypo-perfusion”, the wash out phase is not evaluated for echogenicity
Line 318 to diagnose prostatic tumours
Line 321 multiple planes with image reconstruction
Line 324/325 for examining the prostate in dogs [38] and none of them describe the features of prostatic neoplasia.
Line 327-330 The authors described that the enhancement pattern of prostatic lesions, relative contrast enhancement indices (RCEI), and apparent diffusion coefficient (ADC) values of prostatic lesions may help to detect prostatic adenocarcinoma.
Line 338/339 Improved diagnostic techniques and frequent imaging of the testes in infertility have drawn attention to a significant number of small, solid, and often non-palpable tumours, whose diagnosis and management results significantly problematic for the theriogenologists. – I do not understand this sentence, especially the last part; do you mean: tumours, which are problematic to diagnose and manage by theriogenologists.
Line 344 in longitudinal, transverse
Line 348 a fibrous invagination of the tunica albuginea
Line 375/376 Other findings that may be associated (no comma)
Line 394 3.3. Imaging of penile neoplasia

Author Response

Dear Reviewer,

I very much appreciate the time and effort you’ve put in revising our manuscript and please find attached the latest version on the light of  your comments.

Here follows a point-by-point response and Ido hope that the changes resolve all your concerns.

The section on ovarian tumors should be re-organized, as it is difficult to follow the various imaging and tumor types. To better illustrate the various techniques and imaging characteristics of reproductive tumors, more figures would be beneficiary to the paper.  As requested  the sections of Ovarian tumors has been re-organized and we do hope that is more readable . Images have been added, as requested

General comments:

• The placement of commas should be reviewed, as this also makes reading easier; note the Oxford (serial) comma. I inserted and deleted some commas for better understanding, but not all. As requested  the placement of commas has been reviewed and double checked with the Grammarly Grammar Checker.

  • The headings in the sections are inconsistent. 2. Female continues with 2.1. Reproductive anatomy. In 3. Male, the review starts with 3.1. Imaging of the prostate. For more consistency, there should be a section on reproductive anatomy. As requested we uniformed the headings in the sections.
    • Also in the headings some consecutive headings begin with imaging of the normal uterus or prostate, with a subheading of imaging of neoplasia, others with imaging of vaginal tumors. Imaging of the penis is actually only imaging of penile neoplasia. Again this should be consistent with a normal imaging section and what is possible, followed by neoplasia imaging.
    • In the female section, the headings should be imaging of uterine or ovarian neoplasia, not mass lesions as this would include cysts. In the male section neoplasia was used. As requested we changed and uniformed the terms

  • Figure 2 A and B CT images should be flipped with the spinal cord dorsally. The figure 2 has be oriented in accordance to the reviewer’s comments

Specific comments:

Some suggestions to change some of the text, so it is easier to understand.
Line 17 that in the future they will become (no comma) Revised as requested
Line 37 whilst others provide… Revised as requested
Line 56 are positioned close to the abdominal wall proximally Revised as requested
Line 67 High-resolution probes Revised as requested
Line 78 2.2.1. Imaging of ovarian neoplasia Revised as requested
Line 85 …there are either benign or malignant masses Revised as requested
Line 126/127 by imaging tumour microvascularity Revised as requested
Line 138 Figure 1 Sagittal cut Revised as requested
Line 147 mass with intense contrast enhancement Revised as requested
Line 148 MR; see line 23: MRI – please use consistent abbreviations Revised as requested
Line 151 Imaging of uterine tube neoplasia Revised as requested
Line 152 fallopian tube is not capitalized (see line 153) Revised as requested
Line 162 was described as an ovoid mass – no comma Revised as requested
Line 165 uterus not capitalized Revised as requested
Line 166 MR or MRI? On the light of your comment we revised all MR abbreviation changing them into MRI
Line 169 structures, and, therefore, it cannot be identified…(insert commas) Revised as requested
Line 170 the bitch in a standing position Revised as requested
Line 170/171 after clipping the hair on the ventral abdomen Revised as requested
Line 178 to differentiate these layers depends… Revised as requested
Line 178 During oestrus, (insert comma) Revised as requested
Line 180/181 which represents the mucosal – luminal interface. Revised as requested
Line 181/182 there may be minimal anechoic content (fluid) 1mm wide in the lumen Revised as requested
Line 183 2.4.1. Imaging of uterine neoplasia Revised as requested
Line 190 in veterinary literature Revised as requested
Line 195 tumours of the uterine body Revised as requested
Line 195/196 In half of these dogs, the outer margin of the lesion was well-defined. Revised as requested
Line 197 four dogs and dorsal displacement (no comma) Revised as requested
Line 206 in the mass, giving a mixed (insert comma) Revised as requested
Line 207 reported and thought (no comma) Revised as requested
Line 210 of mixed echogenicity with solid areas (no comma) Revised as requested
Line 211/212 what is meant by “projecting”? and projecting where? What we meant was “ protruding into uterine lumen “ and as requested we have changed

Line 225 as a lipoleiomyoma and also to reveal another (no comma) Revised as requested
Line 219-230 In this section on CT of uterine tumors, it should be noted that there are only 2 case reports. Are there any more? At the time of our submission no other articles were published, but on the light of your comment we found that on the Journal of Small Animal Practice (2021) 62, 293–299 DOI: 10.1111/jsap.13293  in the mid of january the article “CT characteristics of uterine and vaginal mesenchymal tumours in dogs”  come out and for this reason will be included in the references as well

Journal of Small Animal Practice  •  Vol 62  •  April 2021  •  © 2021 British Small Animal Veterinary Association  293

CT characteristics of uterine and

vaginal mesenchymal tumours in dogs

Line 231 Imaging of vaginal neoplasia Revised as requested
Line 234 for visualization of deep proliferative Revised as requested
Line 240 Ultrasonographically, it is extremely difficult Revised as requested
Line 259 vaginal leiomyoma, readily visualized as a contrast-filled defect Revised as requested
Line 280 MRI/MR – inconsistent use of abbreviation Revised as requested
Line 284 One study reported that prostatic - delete reference, already at the end of citation Revised as requested
Line 297 with multiple Revised as requested
Line 299 in these advanced cases. Revised as requested
Line 312 hypo-echogenicity – I believe you meant to say ”hypo-perfusion”, the wash out phase is not evaluated for echogenicityRevised as requested
Line 318 to diagnose prostatic tumours Revised as requested
Line 321 multiple planes with image reconstruction Revised as requested
Line 324/325 for examining the prostate in dogs [38] and none of them describe the features of prostatic neoplasia. Revised as requested
Line 327-330 The authors described that the enhancement pattern of prostatic lesions, relative contrast enhancement indices (RCEI), and apparent diffusion coefficient (ADC) values of prostatic lesions may help to detect prostatic adenocarcinoma. Revised as requested
Line 338/339 Improved diagnostic techniques and frequent imaging of the testes in infertility have drawn attention to a significant number of small, solid, and often non-palpable tumours, whose diagnosis and management results significantly problematic for the theriogenologists. – I do not understand this sentence, especially the last part; do you mean: tumours, which are problematic to diagnose and manage by theriogenologists. Indeed that was the meaning , and  we might add for the clarity of the sentence “ whose delayed diagnosis and treatment could be associated with a poorer survival

Line 344 in longitudinal, transverse Revised as requested
Line 348 a fibrous invagination of the tunica albuginea Revised as requested
Line 375/376 Other findings that may be associated (no comma) Revised as requested
Line 394 3.3. Imaging of penile neoplasia Revised as requested

Reviewer 2 Report

COMMENTS TO AUTHORS

GENERAL COMMENTS

This is a very interesting and well written review dealing with the imaging modalities used for the diagnosis of the reproductive disorders in the dog.

SPECIFIC COMMENTS

Line 48: Please, add comments on the usefulness of the CT and MRI in the imaging of reproductive disorders in the dog.

Lines 58-61: In this sub-chapter you are dealing with the anatomy. Please delete this paragraph or move it to the appropriate chapter in the text.

Lines 81. “When the ovary enlarges ..”. I would suggest “When the ovary is significally enlarged…”

Lines 86: “Radiographic examination has low sensitivity and ….”. I would suggest “Radiographic examination is not sensitive ….” When you say “low sensitivity” you have to bring in the text the percentage of the sensitivity.

Lines 90-93: “Bitches with ovarian tumours……..  echotexture”. Please make the sentences more laconic. For example I would suggest “Ultrasonographically there is an enlarged right or left ovary, with disruption of the normal appearance due to solid/cystic regional or focal lesions. Solid lesion may contain small cysts”.

Lines 93-95: “Ultrasonography……involvement”. This information is well known. Please delete this sentence.

Lines 96-99: “Confirmation that the mass ….Doppler”. Please move these sentences after line 89 and before line 90.

Line 135: Please add “(a, b)” after “Figure 1.” and before “The computed tomography (CT) appearance of a granulosa…”.

Line 166: Replace “includes: with “include”

Lines 213-216: “Ultrasonography has also been described ….. organ can be very challenging”. It should be better for the text uniformity to move these sentences in the beginning of this paragraph.

Lines 203-230: Please, add a couple of ultrasonographic images with different histopathologic types of uterine neoplasia, if any. This paper is dealing with the imaging of the disorders and images are very important for the readers.

Lines 287-314: Please, add a couple of ultrasonographic images with prostatic neoplasia if any.

Lines 357-367: Please, add a couple of ultrasonographic images with testicular neoplasia if any.

Line 369: Please rotate CT images with the spine being in the upper side and note the right (R) and left (L) side of the body in CT images.

Author Response

Dear Reviewer,

I very much appreciate the time and effort you’ve put into your comments and for the very kind words that have been used for commenting the manuscript.

Please find attached copy of the revised version ,on the light of your comments and here follows a point-by-point response to your comments ,hoping that the  the changes resolve all your concerns on the manuscript.

GENERAL COMMENTS 

This is a very interesting and well written review dealing with the imaging modalities used for the diagnosis of the reproductive disorders in the dog.  We really appreciate your, very kind,  comments

SPECIFIC COMMENTS

 Line 48: Please, add comments on the usefulness of the CT and MRI in the imaging of reproductive disorders in the dog.  Many thanks for your comment that will enrich the introduction .The CT characteristics of  canine reproductive  neoplasia can be used to distinguish a malignant from a benign mass. An equally important use of CT in canine reproductive neoplasia will be the use of images in order to create 3D reconstructions,that  may assist the surgeon prior to surgery or during the recovery period. The use of MRI in the canine reproductive neoplasia will provides exquisite detail and contrast of body structures,because this type of imaging is based on chemical composition of the body rather than density.

Lines 58-61: In this sub-chapter you are dealing with the anatomy. Please delete this paragraph or move it to the appropriate chapter in the text. As requested, has been deleted

Lines 81. “When the ovary enlarges ..”. I would suggest “When the ovary is significally enlarged…”. As requested ,we add significantly

Lines 86: “Radiographic examination has low sensitivity and ….”. I would suggest “Radiographic examination is not sensitive ….” When you say “low sensitivity” you have to bring in the text the percentage of the sensitivity.  As requested

Lines 90-93: “Bitches with ovarian tumours……..  echotexture”. Please make the sentences more laconic. For example I would suggest “Ultrasonographically there is an enlarged right or left ovary, with disruption of the normal appearance due to solid/cystic regional or focal lesions. Solid lesion may contain small cysts”. On the light of your precious comment we have rephrased the sentence

Lines 93-95: “Ultrasonography……involvement”. This information is well known. Please delete this sentence.    As requested we deleted the sentence

Lines 96-99: “Confirmation that the mass ….Doppler”. Please move these sentences after line 89 and before line 90. Revised as requested

Line 135: Please add “(a, b)” after “Figure 1.” and before “The computed tomography (CT) appearance of a granulosa…”. Revised as requested

Line 166: Replace “includes: with “include” Revised as requested

 Lines 213-216: “Ultrasonography has also been described ….. organ can be very challenging”. It should be better for the text uniformity to move these sentences in the beginning of this paragraph. The sentence has been moved as requested

Lines 203-230: Please, add a couple of ultrasonographic images with different histopathologic types of uterine neoplasia, if any. This paper is dealing with the imaging of the disorders and images are very important for the readers.   As requested we will add images

Lines 287-314: Please, add a couple of ultrasonographic images with prostatic neoplasia if any. As requested we add ct pre and post contrast , us and contrast of canine prostatic carcinoma

Lines 357-367: Please, add a couple of ultrasonographic images with testicular neoplasia if any. We strongly hope that you do not mind, but we used the contrast ultrasound and b-mode of the canine testicular neoplasia that we believe significant

Line 369: Please rotate CT images with the spine being in the upper side and note the right (R) and left (L) side of the body in CT images.    As requested the images have been rotated and inserted the markers

Round 2

Reviewer 1 Report

Thank you for your revision of the Review Paper - Imaging of Canine Neoplastic Reproductive Disorders. The suggestions made have been correctly addressed and changed or explained. The added images underline the review of an imaging paper, are a benefit for the paper, and are well chosen. Additional references on CT examinations add the latest information.

I still have a small problem with the headings in the female section as they are more inconsistent than in the male section. There is still a mix of normal and neoplasia in the headings (see below).

Specific Comments:

Perhaps you could change the following headings:

Line 259  2.3. Imaging of the uterine tube and uterine tube neoplasia – you continue by describing the normal uterine tube anatomy and that you cannot imagine it.

Or split it into: 2.3. Imaging of the uterine tube, 2.3.1. Imaging of uterine tube neoplasia.

The same is true for:

Line 397  2.5. Imaging of the vagina and vaginal neoplasia

or 2.5. Imaging of the vagina, 2.5.1 Imaging of vaginal neoplasia

Figure 2, line 345 and 357: multiple (no “s”)

Line 394 Interestingly, malignant tumours measured longer than benign forms…

Figure 3 line 480  mixed breed dog (no comma)

Line 481 B-mode

Line 689  3.3.1. Imaging of penile neoplasia (leave out “the”)

Author Response

Dear Reviewer,

Many thanks ,again, for the  time  that you have dedicated and for all the precious comments . 

Please find attached copy of the revised version , and I really appreciate your suggestions that truly improved the review.

best wishes 

Marco 

Round 3

Reviewer 1 Report

Thank you for a very nice review paper. And thank you for the prompt corrections!